# Measuring a Fire. The Story of the January 2019 Fire Told from Measurements at the Warra Supersite, Tasmania

Tim Wardlaw 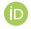

ARC Centre for Forest Value, University of Tasmania, Sandy Bay, Tasmania 7005, Australia; timothy.wardlaw@utas.edu.au; Tel.: +6-14-1836-5329

**Abstract:** Non-stand-replacing wildfires are the most common natural disturbance in the tall eucalypt forests of Tasmania, yet little is known about the conditions under which these fires burn and the effects they have on the forest. A dry lightning storm in January 2019 initiated the Riveaux Road fire. This fire burnt nearly 64,000 ha of land, including tall eucalypt forests at the Warra Supersite. At the Supersite, the passage of the fire was recorded by a suite of instruments measuring weather conditions and fluxes (carbon, water and energy), while a network of permanent plots measured vegetation change. Weather conditions in the lead-up and during the passage of the fire through the Supersite were mild—a moderate forest fire danger index. The passage of the fire through the Supersite caused a short peak in air temperature coinciding with a sharp rise in $CO_2$ emissions. Fine fuels and ground vegetation were consumed but the low intensity fire only scorched the understorey trees, which subsequently died and left the *Eucalyptus obliqua* canopy largely intact. In the aftermath of the fire, there was prolific seedling regeneration, a sustained reduction in leaf area index, and the forest switched from being a carbon sink before the fire to becoming a carbon source during the first post-fire growing season.

**Keywords:** Warra; TERN; *Eucalyptus obliqua*; Riveaux Road fire; fire weather; carbon fluxes; aboveground biomass; regeneration

## 1. Introduction

Under natural conditions, the tall eucalypt forests of south-eastern Australia depend on periodic burning to persist [1,2]. These forests are situated in one of the most fire-prone regions in the world [3] and accumulate biomass to levels that are among the highest globally [4–7]. This high biomass coupled with periodic severe fire weather conditions can result in catastrophic wildfire [8–10].

While these forests can generate some of the most intense fires in the world [11], there are regional differences [2]. In Victoria, fires in tall eucalypt forests (predominantly *E. regnans*) tend to be high intensity and stand-replacing [9]. By comparison, in Tasmania, non-stand-replacing fires predominate [12–14]. Characteristics of intense, stand-replacing wildfires in eucalypt forests of south-eastern Australia are well documented [9,15,16], but there are few reports documenting the characteristics of non-stand-replacing fires, although their fire histories and post-fire regeneration have been documented [12,17].

In a dry lightning storm on the 15 January 2019, more than 2000 lightning strikes ignited 70 fires in remote areas of southern Tasmania. Several of those fires coalesced to form the Riveaux Road fire, which burnt through nearly 64,000 hectares of tall eucalypt forest, buttongrass moorland, silviculturally regenerated eucalypt forest, and eucalypt plantation. The Warra Long-Term Ecological Research site, including the Warra Supersite (https://www.tern.org.au/tern-observatory/tern-ecosystem-processes/warra-tall-eucalypt-supersite, accessed on 13 July 2020), was within the area burnt by the fire. A suite of sensors and a permanent 1 ha plot at the Warra Supersite together with three other 1 ha AusPlots Forests plots [7] provide a rare set of measurements before, during, and after the

fire. Those measurements are used to describe the weather conditions, the fire and the effects of the fire in the tall *E. obliqua* forest.

## 2. Materials and Methods

### 2.1. Study Sites

Data for this study originated from plots and a sensor array of the Terrestrial Ecosystem Research Network (TERN) that were within the fire boundary of the Riveaux Road fire (Figure 1). The core 1 hectare plot of the Warra Supersite (43°5′43″ S; 146°39′19″ E) [18] was used to measure aboveground biomass, tree status (dead/alive), leaf area index, and visual change. An instrumented tower adjacent to the Warra core 1 ha plot was used to measure weather conditions and fluxes of carbon dioxide, water and energy between the forest and atmosphere [19]. A further three 1-ha plots of the Ausplots Forests Network [7] as well as the core 1 ha plot of the Warra Supersite were used to measure seedlings regenerating after the fire.

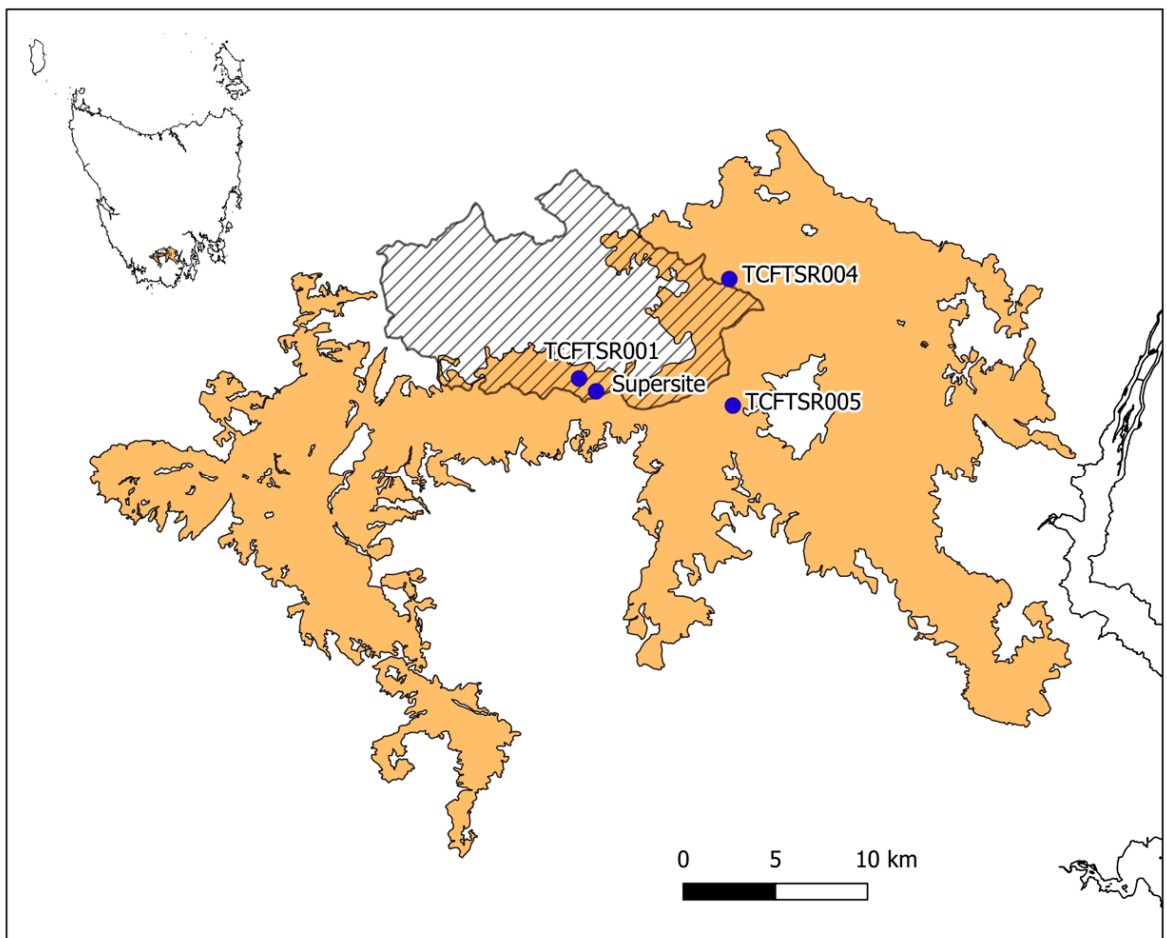

**Figure 1.** Map showing the location of the Warra Long-Term Ecological Research site (hatched) within the mapped extent of area burnt in the Riveaux Road Fire complex (brown shade). The Warra Supersite and the three AusPlots Forests 1 ha plots are shown.

### 2.2. Carbon Fluxes and Meteorology

Measurement of carbon fluxes and meteorology (temperature, relative humidity and wind speed) were done using a suite of sensors mounted on an 80 m tower installed at the Warra Supersite [18,19]. The eddy covariance (EC) system used to measure carbon fluxes consisted of a CPEC200 closed-path EC system (Campbell Scientific, Logan, UT, USA) to measure turbulent fluxes and an AP200 profile system (Campbell Scientific, Logan, UT, USA) to measure temperature profiles and $CO_2$ storage. High frequency (10

Hz) measurements of turbulent fluxes were processed to 30 min averages in a CR3000 datalogger (Campbell Scientific, Logan, UT, USA). High frequency (2 Hz) measurements of temperature profile and storage were processed to 2 min averages sequentially for each of the eight sample heights in a CR1000 datalogger (Campbell Scientific, Logan, UT, USA). Temperature and relative humidity measurements were made with an HMP45 probe (Vaisala, Vantaa, Finland) housed within a naturally aspirated radiation shield. Windspeed measurements were made using a WindSonic4-L 2-D sonic anemometer (Gill Instruments, Lymington, UK) and a CSAT3-B sonic anemometer (Campbell Scientific, Logan, UT, USA). Temperature, relative humidity, and wind speed measurements were processed to 30 min averages and stored in the CR3000 datalogger.

Temperature, relative humidity, and wind speed measurements were used to calculate 30 min values of Forest Fire Danger Index (FFDI) according to [20,21]. Values of Mount's Soil Dryness Index (SDI) to use in the computation of FFDI were provided by the Bureau of Meteorology based on daily weather records measured at the Warra Climate Station, 5.5 km NE of the flux tower.

Flux, storage, and meteorology data were subjected to the standard OzFlux quality control and post-processing through to level 6 as described in [22]. September 1st–December 31st daily records of net ecosystem productivity (NEP) generated at level 6 processing were extracted for each of four years: 2015–2017 and 2019. Records from the September-December period in 2018 were not used because of an extended period of instrument malfunction. Cumulative NEP for the September–December period were calculated for each of the four years. Raw $CO_2$ fluxes measured on the day the fire burnt through the Supersite were accumulated and used to estimate the amount of biomass consumed by the fire. A footprint analysis based on the method of [23] and analytical method developed by [24] was used to calculate and map the footprint of the Warra Flux Tower in the day the fire burnt through the site.

### 2.3. Photograph-Derived Measures

Three types of photograph-derived observations were used: (i) hemispherical photographs used for calculating leaf area index (LAI); (ii) photo-points; and (iii) time-lapse photographs from a fixed camera. The hemispherical and photo-point photographs were each taken from within the core 1 ha plot of the Warra Supersite using a Nikon D7000 camera fitted with either a 10 mm fish-eye lens (hemispherical) or a 24 mm wide-angle lens (photo-points). Both LAI and photo-point methods followed the standard procedures for monitoring vegetation in Supersites documented in [25]. LAI was calculated from the hemispherical photographs using the standard image processing workflow for Supersites as documented in [25]. An average LAI (and the associated 95% confidence interval) for the 1 ha plot (36 photographs) was calculated for each visit—generally twice per year. These data were used to produce a timeseries of average LAI spanning the period December 2015–November 2020. The last four LAI measurements in this time series were after the 2019 fire. Photo-point photographs were taken at approximately annual intervals. The photo-point images received no further processing. The time-lapse photographs were taken using Wingscapes[TM] RBG camera mounted on the top of the Warra flux tower. The camera was programmed to take photographs each hour between 6 a.m. and 6 p.m.

### 2.4. Biomass Measurements within Core 1 ha Plot

Measurement and mapping of all plants >10 cm diameter at breast height (DBH) was done within the core 1 ha plot when the plot was established in 2012–2013. Over-bark diameter measurements were made at breast height (1.3 m above ground level) using a diameter tape. Height measurements were made in one of three ways: (i) hypsometer (Haglöf Vertex) sighting from the ground; (ii) identified maximum height within LiDAR point-clouds of delineated tree crowns; (iii) predicted from species-specific least-squares regression relationships of DBH versus measured height (using methods i or ii). Ground-

measured heights using a hypsometer were concentrated in understorey trees and manfern, *Dicksonia antarctica* (Table 1).

**Table 1.** Number of each species of trees, shrubs, and ferns in the core 1 ha plot at the Warra Supersite that were heighted by each of the three methods used to measure height.

| Taxa | Measured by Hypsometer | Measured by LiDAR | Predicted from Regression |
|---|---|---|---|
| *Acacia melanoxylon* | 23 | | 55 |
| *Atherosperma moschatum* | 11 | | 8 |
| *Dicksonia antarctica* | 255 | | |
| *Eucalyptus obliqua* | 6 | 39 | 61 |
| *Nothofagus cunninghamii* | 21 | | 79 |
| *Pomaderris apetala* | 5 | | 95 |
| Other | | | 12 |

LiDAR was used to measure heights of the taller *E. obliqua*, which formed the upper canopy of the forest. The LiDAR data were captured in summer 2009–2010 by AAM (https://www.aamgroup.com/, accessed on 2 November 2020) using an Optech Gemini discrete-return scanner with a minimum point density of 20 points/m$^2$ [26]. Vegetation height was calculated with reference to a one metre resolution digital terrain model and plotted as a colour-coded canopy map on which individual tree crowns were manually delineated. The canopy map of delineated tree crowns was matched to numbered trees in the field. The maximum height in each delineated tree crown was extracted from the LiDAR returns. For each taxon that had paired DBH and height measurements, a plot of DBH versus measured height was done and a least-squares regression was fitted to those data using the "Comparison of regression lines" routine in Statgraphics (Statgraphics Technologies Inc., Virginia). The fitted regression models for each species (Appendix A Table A1) were used to predict height values for the remaining trees of the species in the core 1 ha plot based on their measured DBH. One further regression was fitted by combining DBH and measured height of the four main understorey tree species (*Acacia, Nothofagus, Atherosperma, Pomaderris*). This regression model was used to predict heights of the small number of individuals of five other species (*Eucryphia lucida, Pittosporum bicolor, Phyllocladus aspleniifolius, Melaleuca squarrosa, Monotoca glauca*).

DBH and height measurements were used to calculate the entire stem volume of each tree. Species-specific equations obtained from published literature were used (Appendix A Table A2). Entire stem volumes were converted to biomass using published values of basic density for individual species (Appendix A Table A3). Entire stem biomass was converted to estimates of total above-ground tree biomass using an expansion factor of 1.58 for *E. obliqua* (=1/0.632 as stem volume contains 63.2% of total aboveground biomass in [27]) and 1.46 for all other species based on [28].

All plants censused in the 2012–2013 measurement of the core 1 ha plot were assessed in a remeasurement of the plot in November 2019—ten months after the fire. This was considered sufficient time to determine whether plants had been killed by the fire; recovered after crown scorch; or were unaffected by the fire. The status of standing plants (dead or alive) in 2012–2013 and the 2019 censuses were compared and tabulated to show the number of plants killed/surviving by the 2019 fire and the biomass they represent.

*2.5. Small Coarse Woody Debris (CWD)*

Small (10–40 cm diameter) CWD was assessed on two occasions—June 2015, prior to the fire, and February 2021, after the fire. The line intersect transect method [29] was used. Four transects, each 50 m long, were sampled. Each transect originated from one of the four sides of the core 1 ha plot of the Warra Supersite and extended, perpendicularly, 50 m into the plot. The same transects were used on both occasions. Each piece of CWD encountered along the transects that was within 10–40 cm diameter where it intersected

with the transect was assessed. For each piece of CWD within this size range the following attributes were assessed: distance along transect; diameter at its point of intersection with the transect (measured in the horizontal and vertical planes using calipers); decay class using the 5-point classification scale described in [30]; host species (if known), origin (stem, branch or root); and the presence of charring. The volume of each piece of CWD was calculated according to the formula of [29]; converted to weight using decay-class density average values given in [31]; and finally converted to weight of carbon by multiplying by 0.497, the ratio of carbon in *E. obliqua* at Warra according to the study of [27].

Pieces of CWD assessed in 2015 that were also present in 2021 (i.e., persisted after the 2019 fire) were determined as occurring at the same distance along the transect and were of similar diameter on both occasions. Data were tabulated to report CWD biomass potentially consumed by the 2019 fire (present in 2015 but not 2021); CWD added between the 2015 assessment and the fire (CWD not present in 2015 and charred); and CWD added after the fire (CWD not present in 2015 and not charred).

### 2.6. Post-Fire Seedling Regeneration

Seedling density was measured 10 months after the fire in four 1 ha Ausplots Forest plots [plots TCFTSR001,2,4 and 5 of Supporting information S2 in 7] that had been burnt in the January 2019 fire. A modified point intersect transect method [32] was used. A 10 × 10 cm quadrat was sampled at each of 1010 points in each of the four plots. Each quadrat was centered by a laser point from a downward pointing laser pointer mounted on a vertical staff. The number of seedlings within each quadrat was counted. Seedling counts were partitioned by taxon where the taxon could be determined: *Eucalyptus obliqua*, *Acacia* species (*A. melanoxylon* and *A. verticellata*), *Pomaderris apetala*, and "other species". The average number of seedlings per quadrat was calculated and converted to seedlings per hectare by multiplying by one million.

## 3. Results

### 3.1. Fire Weather Conditions

The Riveaux Road fire complex burnt nearly 64,000 ha over a three-week period between the 15th January and the 6th February. About half of that area was in tall eucalypt forest, which included the Warra Supersite. The Warra Supersite recorded weather conditions during the fire from the 15th January to the 28th January when the fire burnt through the site and power was lost.

The forests were quite moist in the period leading up to the fire. A heavy rain event on 20th December 2018 resulted in the soil dryness index dropping almost back to zero (Figure 2e). Between 20th December 2018 and 7th February 2019, the soil dryness index steadily increased and had reached 55 mm by the time the fires started on 15th January and peaked at 99 mm on 6th February—the day before significant rainfall.

High relative humidity, mild temperatures and moderate winds were measured at the Warra Supersite for much of the period between 15th and 28th January (Figure 2a–c). Consequently, the Forest Fire Danger Index remained in the moderate range for much of the period (Figure 2d). Significant peaks in the FFDI on 21st January and 25th January corresponded with the initial run of the Riveaux Road fire and the Clearwater Creek fire (later merging with the Riveaux Road fire) jumping across the Huon River, respectively.

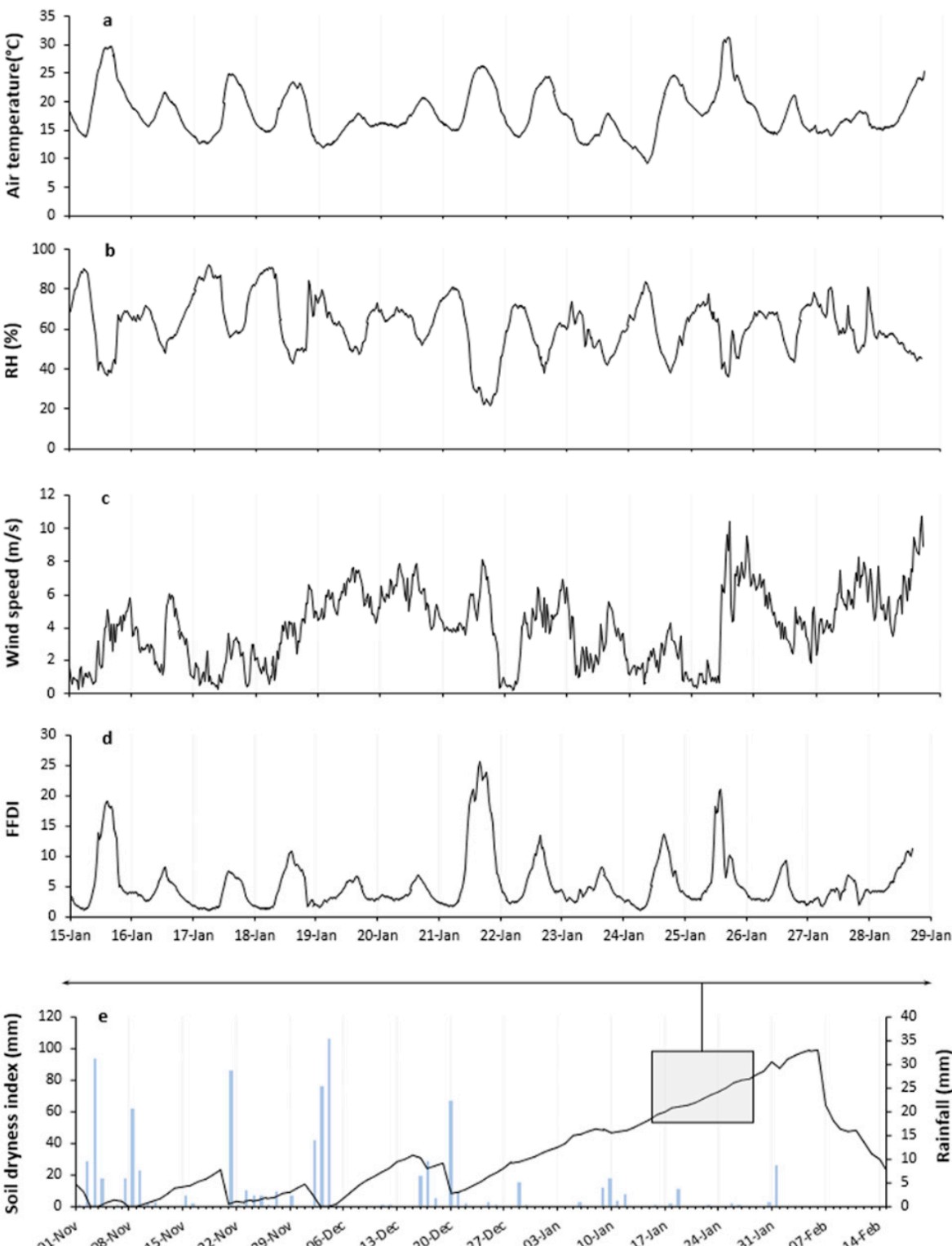

**Figure 2.** Weather conditions measured at the top of the flux tower at Warra Supersite (**a–d**) and the Bureau of Meteorology's Warra Climate Station (**e**). Timeseries from measurements at the Warra Supersite show 30 min averages for the period 15th–28th January 2019 of: (**a**) air temperature; (**b**) relative humidity (RH); (**c**) wind speed; and (**d**) Forest Fire Danger Index. Rainfall and Mount soil dryness index from the Warra Climate Station are shown as a timeseries of daily measurements (**e**).

### 3.2. Measurements as the Fire Burnt through the Warra Supersite

Thermocouples in the profile system installed on the flux tower measured a sharp rise in temperature just after 17:00 on the 28th January (Figure 3a). Temperature peaked at 70.7 °C at a height of 16 m. The temperature quickly dropped from that peak but fluctuated at above-ambient temperature for another 5 min until power was lost at 17:08. Photographs from a time-lapse camera mounted on the top of the tower indicated the fire did not continue to burn once the fire front has passed through the site—there was no evidence of smoke generated by continued smouldering from the forest floor on the day following the fire.

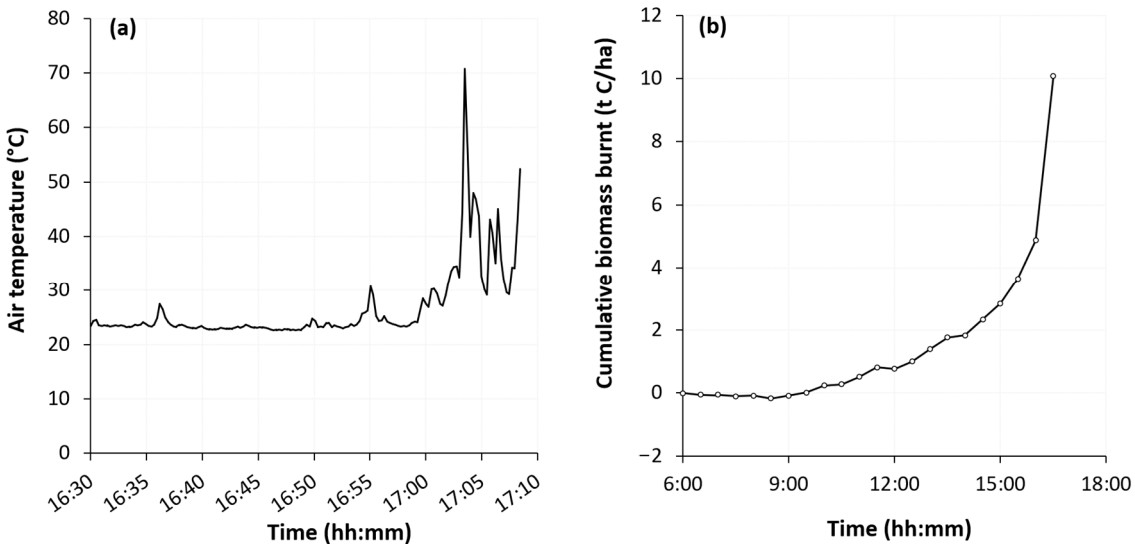

**Figure 3.** Measurements made as the fire burnt through the Warra Supersite: (**a**) air temperature measured at a height of 16 m; (**b**) cumulative biomass burnt.

From 09:00 on the 28th January the EC instruments at the top of the tower measured a progressive release of $CO_2$ from the forest (Figure 3b). By 15:00 nearly 3 t of carbon had been released from the forest (as $CO_2$). After that there was a sharp increase in the amount of $CO_2$ released by the forest with over 5 t of carbon being released as $CO_2$ in the final half hour period before the loss of power to the site. In total, the eddy-covariance system measured the loss of 10 t C/ha throughout the day of the 28th January as the fire burnt through the site, up until the time power was lost. The footprint for the tower on the day the fire burnt through the site shows the core 1 ha plot and immediate environs (approximately 350 m) to the west was contributing a high proportion of the measured fluxes (Figure 4)

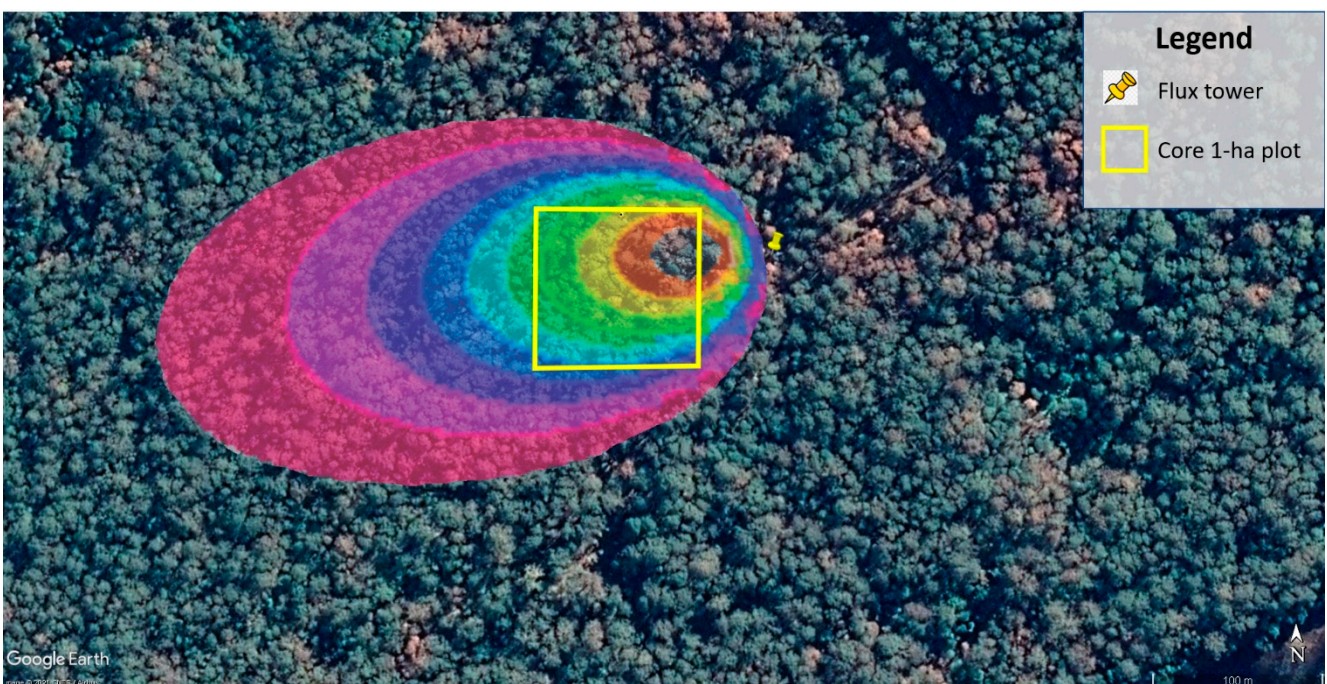

**Figure 4.** Google Earth™ map of the Warra Supersite showing the location of the flux tower and core 1 ha plot relative the tower footprint for the day the fire burnt through the site on 28th January 2019. The footprint colours reflect the cumulative contribution to the total fluxes in decile steps from 10% (inner uncoloured area) to 90% (outer, pink-coloured area).

### 3.3. Changes to the Existing Forest

Before and after photographs from photo-points in the core 1 ha plot of the Supersite (Figure 5) showed near-complete combustion of the ground vegetation and litter, but only surface charring of larger diameter coarse woody debris (Figure 5b). Foliage in trees occupying the mid and upper strata of the understorey was generally not burnt but was scorched from radiant heat. There was some patchy scorching of the foliage of the overstorey eucalypts but generally the overstorey canopy remained green (Figure 6). Despite the observation of a largely intact overstorey canopy, the leaf area index (LAI), measured four months after the fire, had dropped by one third from 5.7 to 3.9 $m^2/m^2$ (Figure 7). Eighteen months later the LAI showed little recovery towards pre-fire levels.

There was a large turn-over of small diameter (10–40 cm) CWD on the forest floor before and after the 2019 fire (Table 2). Three-quarters of the small CWD that was present in 2015 had disappeared by 2021, either through combustion or decomposition. The remaining 25% of the CWD present in 2015 persisted after the 2019 fire to be recorded in 2021. Most (55%) of the CWD present in 2021 was uncharred and was assumed to have been new CWD additions following the 2019 fire. Two thirds of the remaining CWD present in 2021 was not present in 2015, and was charred, and so was assumed to have been added between 2015 and the 2019 fire. The remainder was CWD that was present in 2015. This component of CWD had lost nearly 1 t/ha of its 2015 biomass carbon.

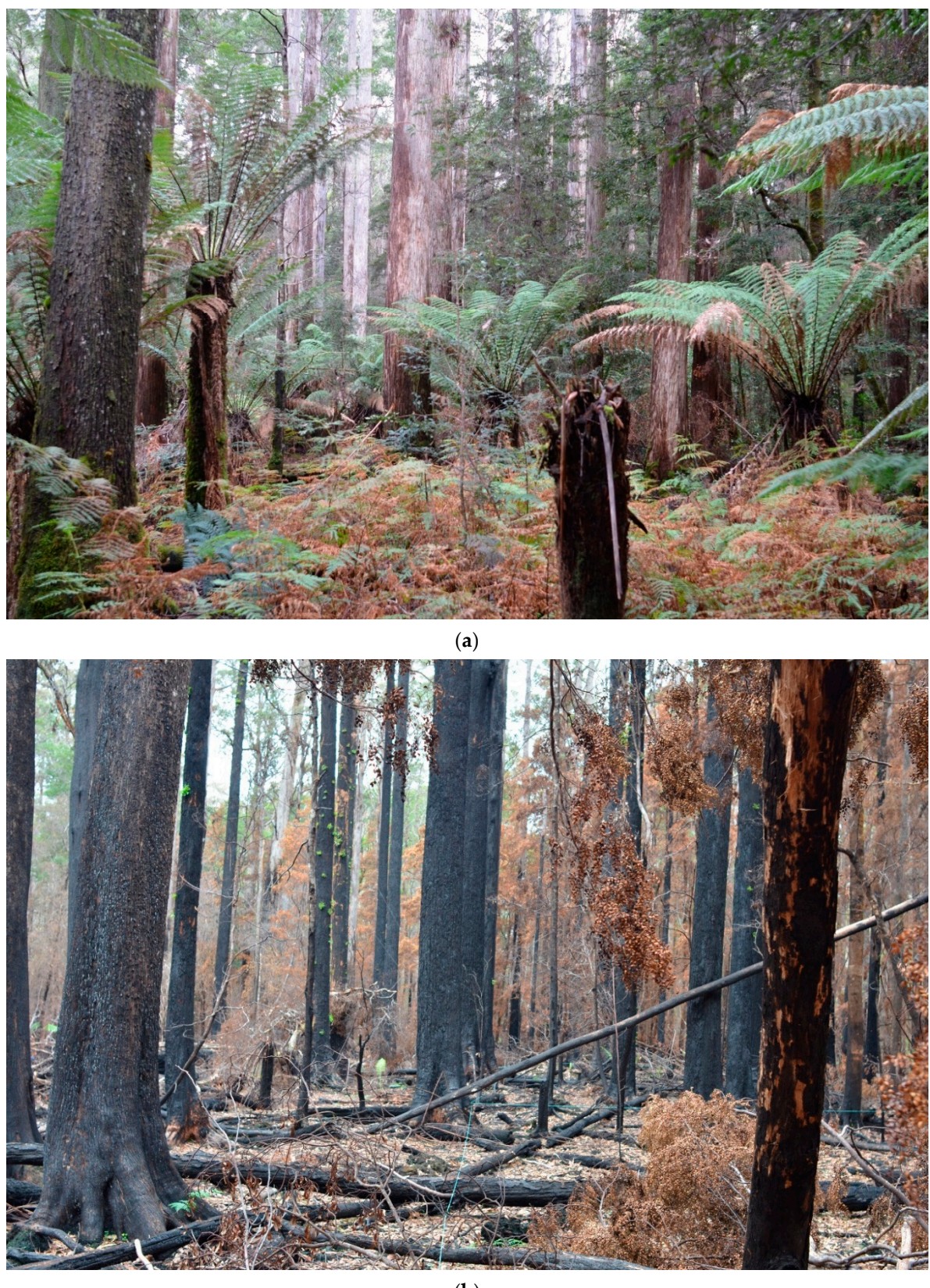

(**a**)

(**b**)

**Figure 5.** Forest scene looking to the south from photo-point 100, 100 (x, y co-ordinates in metres) in the core 1 ha plot of the Warra Supersite, (**a**) before and (**b**) after the 2019 fire showing near-complete combustion of the ground layer and foliage scorch of the mid-layer understorey.

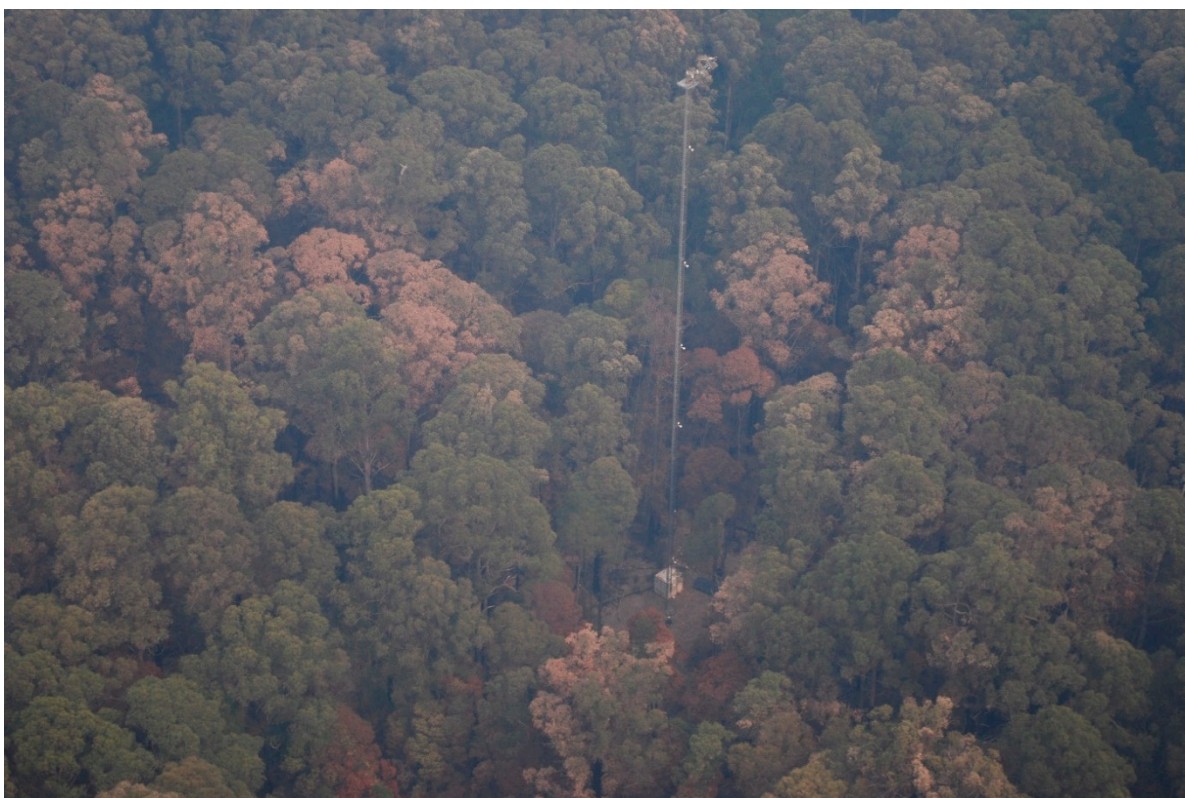

**Figure 6.** Aerial view of the Warra Supersite nine days after the fire passed through the forest showing patchy foliage scorch in a largely still green eucalypt overstorey canopy.

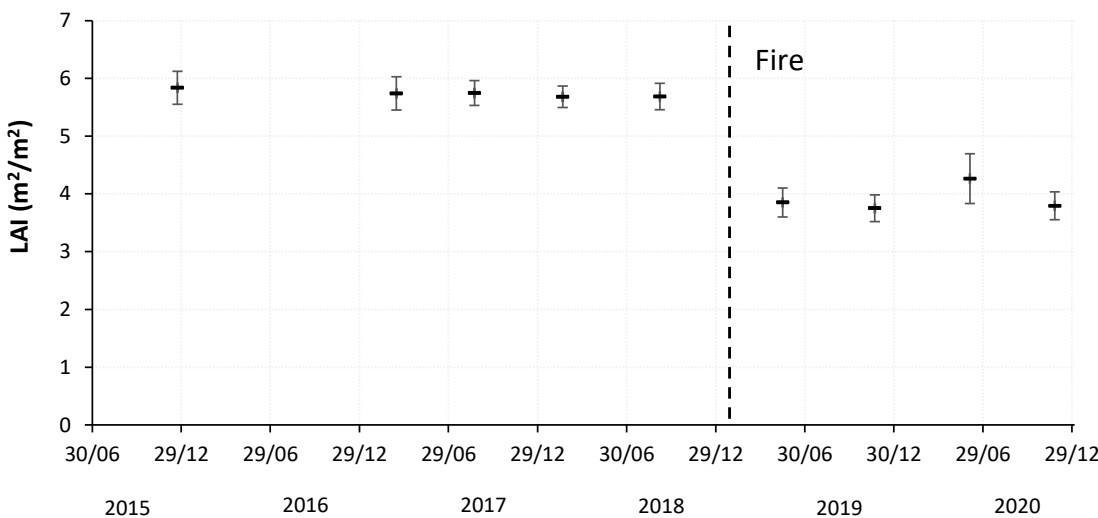

**Figure 7.** Average leaf area index (LAI) measured within the core 1 ha plot at the Warra Supersite between 2015 and 2020; 95% confidence intervals are shown.

**Table 2.** Census of small (10–40 cm diameter) CWD measured on two occasions (2015 and 2021) using line intersect transect surveys in the core 1 ha plot in the Warra Supersite. Percentages of column total are shown in parentheses.

| CWD Demographic | BIOMASS Carbon (t/ha) | |
| --- | --- | --- |
| | **2015** | **2021** |
| Present in 2015 and persisting to 2021 | 3.5 (24) | 2.6 (14) |
| Present in 2015 BUT not persisting to 2021 | 10.9 (76) | |
| Added between 2015 and 2019 fire (charred) | | 5.6 (31) |
| Added after 2019 fire (not charred) | | 10.0 (55) |
| Total | 14.4 | 18.2 |

There were sharp differences in survival among plant species (>10 cm DBH) on the core 1 ha plot. *Eucalyptus obliqua* and *Dicksonia antarctica* both had high survivorship (88 and 75%, respectively). However, *Acacia melanoxylon* and other understorey tree species, including *Pomaderris apetala, Nothofagus cunninghamii*, and *Atherosperma moschatum* all suffered more than 80% mortality (Figure 8). Despite the high mortality of understorey plants, the fire caused a relatively small (28% = 216 t/ha) reduction in the total above ground biomass contained in living plants. This was because *E. obliqua*, which had high survivorship, contained a disproportionately large amount (72%) of the aboveground biomass (Figure 9).

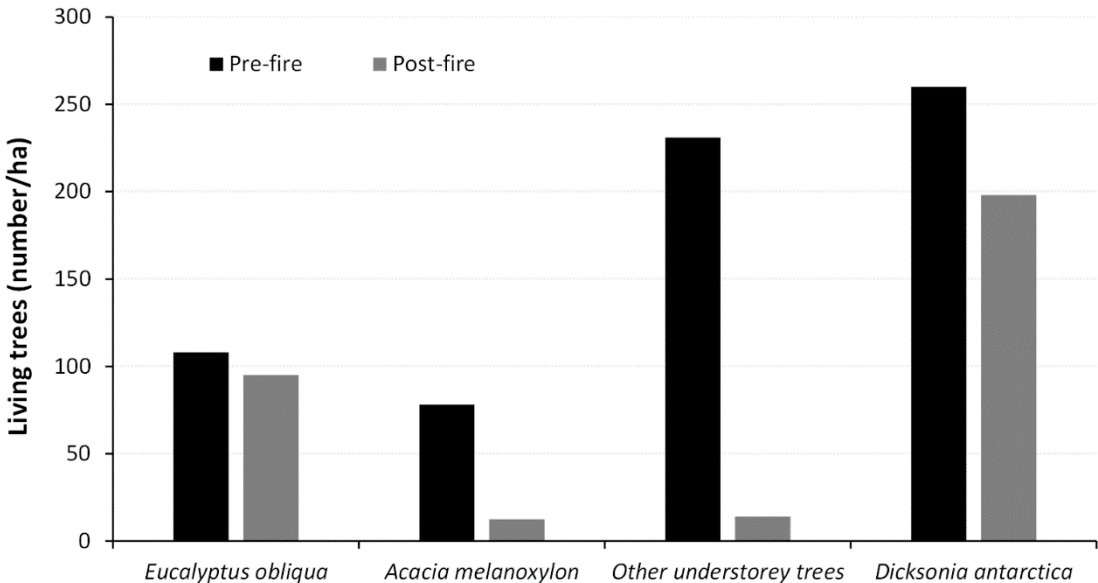

**Figure 8.** Census of living plants > 10 cm DBH, aggregated into major groups, on the core 1 ha plot of the Warra Supersite before and after the 2019 Riveaux Road fire.

Prior to the 2019 fire, the tall *E. obliqua* forest at Warra typically removed between 5 and 10 t of $CO_2$ from the atmosphere during the main part of the growing season running from 1st September to 31st December each year (Figure 10). In the first growing season after the fire, instead of removing 5–10 t of $CO_2$, the forest released more than 5 t of $CO_2$ back into the atmosphere (Figure 10). This switch from a carbon sink to a carbon source was mainly driven by a strong decline in gross primary productivity (GPP) rather than an increase in respiration (data not shown).

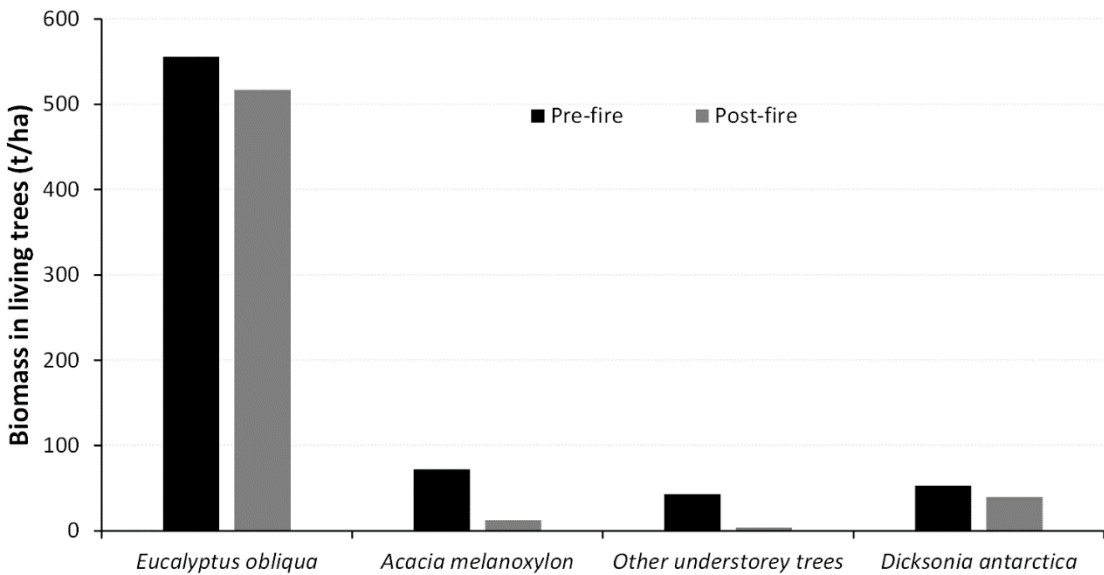

**Figure 9.** Biomass contained in living trees >10 cm DBH, aggregated into the major groups, on the core 1 ha plot of the Warra Supersite before and after the 2019 Riveaux Road fire.

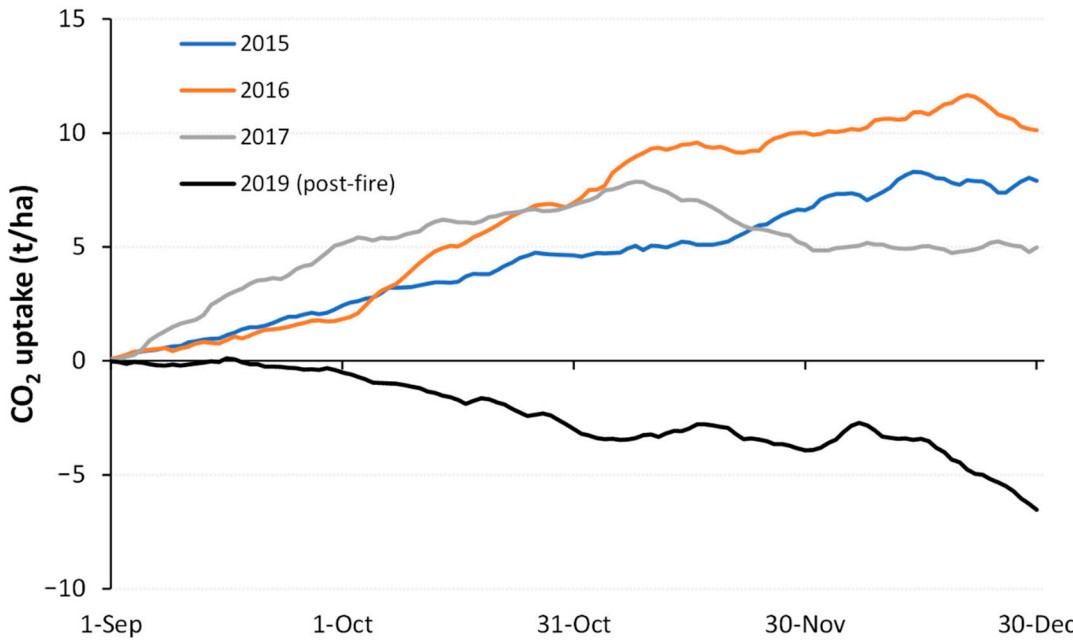

**Figure 10.** Net seasonal (1 September to 31 December) $CO_2$ uptake (+ve) or loss (−ve) by the forest at the Warra Supersite in each of four years. Sensor malfunction prevented $CO_2$ flux measurements for much of 2018.

### 3.4. New Cohort of Plants Regenerating after the Fire

Seedling regeneration appeared at the Warra Supersite soon after the fire. A limited survey done 135 days after the fire in the core 1 ha plot estimated nearly 1.5 million seedlings per hectare. *E. obliqua* and *Acacia* seedlings, which could be identified at that young age, achieved densities of over 500,000 and 300,000 seedlings per hectare, respectively. Time-lapse photographs taken from the top of the Warra flux tower recorded a heavy flowering event of the *E. obliqua* at the Supersite in February and March 2017—two years before the fire. That seed crop would have reached maturity by the time the fire burnt the site. *A. melanoxylon* regularly flowers in late winter at the Supersite and would thus have developed a large seedbank in the soil [33].

More extensive seedling density surveys of four 1 ha Ausplot Forests plots [7] done in November—240 days after the fire—estimated an average of 450,000 seedlings per hectare (Figure 11a). *E. obliqua* and *Pomaderris apetala* each had densities of approximately 45,000 seedlings per hectare and there were 114,000 *Acacia* seedlings per hectare. Seedlings that could not be identified with a taxon contributed to more than half of the seedlings recorded in the November surveys. The November surveys did record similar seedling densities as the limited survey done 135 days after the fire in the core 1 ha plot at the Supersite. However, when the data were restricted to the survey points common to both surveys, a substantial reduction in the density of *E. obliqua* and *Acacia* seedlings occurred in the interval between the 135- and 240-day post-fire surveys (Figure 11b).

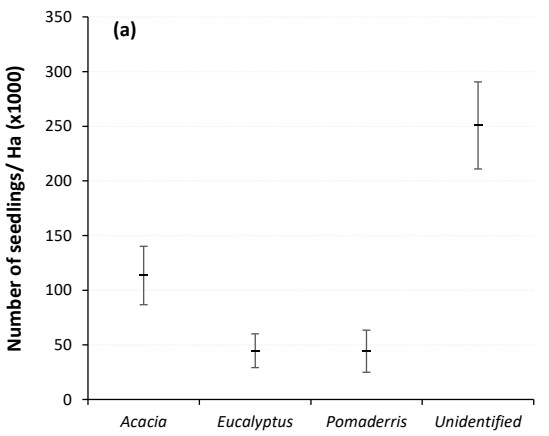 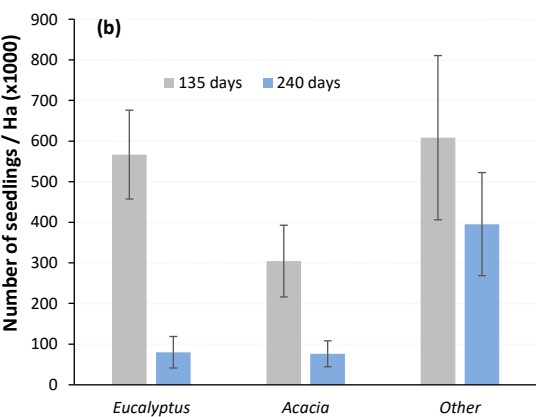

**Figure 11.** Census of the average number of seedlings: (**a**) in the four 1 ha AusPlots Forests that were burnt in the 2019 Riveaux Road fire; (**b**) 135 and 240 days after the fire in a subset of points in the core 1 ha plot at the Warra Supersite. Standard error bars are shown.

## 4. Discussion

This study has described a non-stand-replacing wildfire in a Tasmanian tall eucalypt forest with unprecedented detail using a diverse suite of before-, during- and after-fire measurements. These measurements describe a low intensity fire burning under mild weather conditions with a short residence time that consumed fine fuels on the ground and some small diameter CWD; scorched the foliage of most understorey trees but few of the overstorey eucalypts; generated prolific seedling regeneration; and triggered a switch from the stand taking-up carbon during the September-December growing period prior to the fire to the stand losing carbon for the same period in, at least, the first year after the fire.

EC instruments on the flux tower at Warra measured the release of $CO_2$ equivalent to 10 t/ha of biomass carbon during the day the fire burnt through the site. Most of that $CO_2$ was released during the passage of the fire front coinciding with an abrupt rise in air temperature. Such measurements do not usually get published because the meteorological conditions fail to satisfy the requirements needed for the EC technique and the data from Warra during the fire were no exception—all raw 30 min averages of $CO_2$ fluxes during the fire failed the QA range tests used in PyFluxPro [22]. Despite this, the raw fluxes are of interest when compared with biomass losses from combustion measured by other means. [34] measured 20 and 7.5 t per hectare in fine fuels in ground litter and fern fronds, respectively, at the Supersite pre-fire. [35] measured biomass in combustible fuels in the ground layer of about 20 t/ha in several nearby tall eucalypt forests prior to their harvest. These fuel types were completely combusted, or nearly so, by the 2019 fire at the Supersite (Figure 5a,b), and the measured 27.5 t/ha biomass would therefore equate to the combustion of about 13–14 t C/ha. A similar amount of small diameter CWD could also have been combusted, although the CWD survey was unable to disentangle C-losses from combustion with losses from decomposition between the 2015–2021 CWD assessments. The combined C-losses in fine fuels/ground vegetation and small diameter CWD was

about double the losses measured by EC, but the EC measurement was truncated soon after the main fire front passed through the site.

The moderate weather conditions, as reflected in the FFDI remaining in the moderate range, was reflected in conditions during the passage of the fire through the flux site. Air temperatures measured by the profile system on the flux tower during the passage of the fire were much lower than those measured in test fires used to evaluate fire-fighter safety equipment [36]. Around half of the small diameter CWD that was present before the fire persisted after the fire, consistent with a woody fuel consumption in a low (<600 kW/m$^2$) intensity fire [37]. However, patchy crown scorch of the 50 m-tall *E. obliqua* canopy suggests locally much higher fire intensity, e.g., ≈6000 kW/m$^2$ based on equation 11 in [38]. Such localized crown scorch seemed to be associated with linear openings in the canopy such as roads or, in the case of the flux tower guyline alleys (Figure 6), where wind may penetrate to the forest floor more easily.

The 2019 Riveaux Road fire complex had many commonalities with the January 2016 fire in the Lake MacKenzie area of northern Tasmania. Both fires were ignited from lightning [39] and fire weather conditions were mild [34]. Those two fires were much less severe than the first two days (3 and 4 January 2013) of the Dunalley Fire [10], which burnt during a record heatwave [40]. Accordingly, we expected to see similarities between the Riveaux Road and Lake MacKenzie fires in their impact in comparable areas of tall eucalypt forests. High survivorship of the overstorey eucalypts and the treeferns (*D. antarctica*) recorded at the Warra Supersite were also recorded at the Lake MacKenzie AusPlot Forests plot [41]. In contrast to [41], however, where most understorey trees survived, the Riveaux Road fire resulted in the death (from crown scorch) of the majority of the understorey trees. As the weather conditions were similar when the fires burnt through the forest plots [34], the contrasting effects on the understorey trees may have been due to differences in fuels—Warra Supersite had higher fine and coarse fuel loads and higher density of manferns than Lake MacKenzie [34] and thus may have supported a higher intensity of ground fire.

Many of the understorey species killed by the fire at Warra were regenerating in the new stand. *Pomaderris apetala* and *Acacia melanoxylon*, which were the second and fourth main contributors to biomass in the original stand were both plentiful in the seedling regeneration. Shrubs such *Acacia verticellata* and *Zieria arborescens* were also common in the seedling regeneration but were very rare in the original forest. They are likely to have been early colonisers of the original stand but had largely disappeared by the time the core 1 ha plot was established and measured.

The high survivorship of the existing stand of *E. obliqua* after the 2019 fire contrasts with the 1898 fire—the last fire to burn the forest at the Warra Supersite. At the Supersite, only a small proportion of the stand existing in 1898 survived that fire. This assertion is based on the numerical dominance of trees regenerating from the 1898 fire compared with surviving trees of the existing stand, i.e., "mature" trees in the current forest, which represented only four of the 108 eucalypts in the core 1 ha plot when the 2019 fire occurred. The 1898 fire likely burnt under more severe fire weather conditions, as southern Tasmania was in the grip of record summer (Dec–Feb) drought conditions at the time (Bureau of Meteorology records for Cape Bruny Lighthouse) coinciding with above-average temperatures (Bureau of Meteorology records for Elderslie Rd, Hobart). The fate of the crop of *E. obliqua* seedling regeneration after the 2019 fire is difficult to predict. While LAI decreased by a third after the fire, it was still high (approximately 4 m$^2$/m$^2$) relative to many Australian eucalypt forests [42]. In a comparable fire that killed the understorey but left the eucalypt overstorey intact, [43] found *E. regnans* seedling regeneration did not persist beyond two years after the fire. Similarly, [44] documented low stocking and growth of *E. obliqua* seedlings, three years post-harvest, in the small gaps created by single tree/small group harvesting in a mature *E. obliqua* stand.

The fate of the 2019 cohort of eucalypt seedlings could have a significant effect on the future level of carbon stored by the forest. This is because eucalypts dominate the

productivity of the forests in which they grow based on their dominant contribution to aboveground biomass shown here and elsewhere [7,45]. EC measurements indicated the forest had switched from a strong carbon sink during the Sept-Dec growth period prior to the 2019 fire to a strong carbon source for the same period after the fire. This is a typical response of the carbon dynamics of forests to fires [46–49]. However, the recovery time, post-fire, for a forest to return to being a carbon sink again is variable. Rapid recovery (two years post-fire) to a net sink was measured in a eucalypt mallee ecosystem where both respiration and GPP were depressed post-fire, but GPP recovered more quickly [47]. Other studies have measured much slower recovery, which authors attribute to the ongoing respiration of fire-killed trees, particularly after they fall to the ground where decomposition is accelerated [48,49]. The initial change from a pre-fire carbon sink to a post-fire carbon source measured in the forest at Warra was due to the reduction in GPP as would be expected from a 30% reduction in LAI. While most of the aboveground carbon at Warra still resided within living trees post-fire, there was nonetheless more than 200 t/ha in fire-killed trees. Respiration will be expected to increase substantially as this biomass decomposes and may keep the forest as a net carbon source for an extended period as has been measured elsewhere [48]. However, GPP may increase above levels measured pre-fire if the 2019 cohort of eucalypt seedlings are able to persist and actively grow. The balance between possible enhanced GPP post-fire and elevated respiration from the decomposition of fire-killed trees is of great interest in these tall eucalypt forests. This is because of their substantial contribution to carbon stored in Tasmania's forests [50]; the role of the land-use, land-use change, and the forestry sector in offsetting emissions in Tasmania's greenhouse gas accounts [51]; and the predicted increase in conditions favouring fires with climate change [52].

## 5. Conclusions

Obtaining measurements during unplanned wildfires is challenging and often restricted to a narrow range of parameters being measured. In this case study, a diverse suite of sensor and plot-based measurements collected on an ongoing basis at Warra was used opportunistically to provide before, during and after fire measurements. Using a mix of plot-based and sensor measurements enabled links between changes in the biological components and the biophysical processes to be identified. Exemplifying this was an important finding of a strong stratification of fire impacts: the eucalypt canopy was left largely intact; the understorey tree/shrub strata remained structurally intact but were almost completely killed; the ground layer was consumed but ferns mostly survived to resprout. The stratification of impacts created contrasts in the carbon dynamics of the forest. Most of the aboveground carbon remained in living eucalypt trees, but the forest switched from being a $CO_2$ sink during the growing season before the fire to a $CO_2$ source after the fire. How long the forest remains a $CO_2$ source will depend on the balance between the respiration of the fire-killed understorey versus the photosynthesis of the dense crop of seedling regeneration. This will only be answered by ongoing monitoring at the Supersite.

This case study provides a template of the opportunities afforded by co-opting long-term monitoring infrastructure to advance current and future research programs seeking to better understand and manage fire.

**Funding:** Data used in this study was generated from infrastructure of the Terrestrial Ecosystem Research Network through funding provided by the National Collaborative Research Infrastructure Scheme of the Commonwealth of Australia.

**Institutional Review Board Statement:** Not applicable.

**Informed Consent Statement:** Not applicable.

**Data Availability Statement:** Data used in this study is freely available from the Data Portal of the Terrestrial Ecosystem Research Network (https://www.tern.org.au/data/, Accessed on 8 March 2021). At the time of writing the TERN data repository was undergoing a major rebuild and datasets used in the study are not yet discoverable via the public data portal. Flux data

(Warra_yyyy_L6.nc); LAI (https://bioimages.tern.org.au/data/lai/wrra/default); Photopoint ( https://bioimages.tern.org.au/data/photopoint/wrra/default); biomass (native file lodged with TERN – wrra_core_1ha_vegetation_biomass_2013-19.csv); point-intersect survey of seedling density (native file lodged with TERN – wrra_core_1ha_vegetation_point_intersect_seedling_2019.csv).

**Acknowledgments:** The data used in this paper was provided using research infrastructure of the Terrestrial Ecosystem Research Network. Alison Phillips (Sustainable Timber Tasmania) has provided ongoing technical support for the operation of the infrastructure and resultant datasets at the Warra Supersite. Ben Sparrow, Emrys Leitch, Michael Starkey, Tamara Potter, Nicki Francis, Lachlan Pink and Anton Steketee (Adelaide University) undertook the point intercept surveys of three of the AusPlots Forests (TCFTSR001, 4 and 5) and assisted with the 2019 remeasurement of plants >10 cm DBH in the core 1 ha plot of the Warra Supersite. Cacelia Ewenz (TERN Ecosystems Processes Central Node) did the footprint analysis. The Incident Management Team for the Riveaux Road Fire made a great contribution in limiting damage to the infrastructure at the Warra Supersite and acting quickly to restore safe access to the site after the fire.

**Conflicts of Interest:** The author declares no conflict of interest.

## Appendix A. Allometric Equations Used for Computation of Aboveground Biomass in the Core 1 ha Plot of the Warra Supersite

**Table A1.** Least squares regression models to predict tree height as a function of DBH (diameter at breast height) measurements for taxa present within the core 1 ha plot at the Warra Supersite.

| Taxon | Regression Model | Statistics |
|---|---|---|
| *Eucalyptus obliqua* | Ht = 1/(0.0158539 + 0.423019/DBH) | $F_{1,43}$ = 51.7; MSE = 6.34 $\times$ $10^{-6}$; r = 0.739 |
| *Acacia melanoxylon* | Ht = 1/(0.0138618 + 0.625844/DBH) | $F_{1,22}$ = 31.1; MSE = 2.18 $\times$ $10^{-5}$; r = 0.765 |
| *Nothofagus cunninghamii* | Ht = 34.1307 − 228.262/DBH | $F_{1,19}$ = 60.9; MSE = 8.00; r = −0.873 |
| *Atherosperma moschatum* | Ht = 1/(0.01889 + 0.655262/DBH) | $F_{1,9}$ = 39.2; MSE = 4.21 $\times 10^{-5}$; r = 0.902 |
| *Pomaderris apetala* | Ht = 24.8721 − 155.257/DBH | $F_{1,3}$ = 1.1; MSE = 9.57; r = −0.517 |
| Other species | Ht = 1/(0.0126758 + 0.734503/DBH | $F_{1,58}$ = 164.4; MSE = 8.83 $\times$ $10^{-5}$; r = 0.860 |

**Table A2.** Equations used to calculate entire stem volume (ESV) for measured tree height (Ht) and diameter at breast height (dbh) of taxa in the core 1 ha plot at the Warra Supersite.

| Taxon | Stem Volume Model | Reference |
|---|---|---|
| *Acacia melanoxylon* | Ln (ESV) = −8.359 + 2.265 ln(dbh) | [53] |
| *Nothofagus cunninghamii* | ESV = 0.00125338 $\times$ ((1 − $e^{(-0.0380063 \times Ht^{0.75})}$) $\times$ $dbh^2$) | [54] |
| *Eucalyptus obliqua* | ESV = $e^{(-6.748851 - 0.0035614 \times dbh + 0.1893998 \times Ht - 0.0013634 \times Ht2 + 2.207765 \times (dbh/ht) - 0.2535472 \times (dbh/Ht)2)}$ | R.Musk pers. comm. (from Forestry Tasmania inventory data) |
| *Eucryphia lucida* | ESV = 0.00107074 $\times$ ((1 − $e^{(-0.0618753 \times Ht^{0.856899})}$) $\times$ $dbh^2$) | [54] |
| *Phyllocladus aspleniifolius* | ESV = 0.00108502 $\times$ ((1 − $e^{(-0.0562587 \times Ht^{0.871411})}$) $\times$ $DBH^2$) | [54] |
| Other species | 1.3 m-top = conic volume; 0–1.3 m cylindric volume | Default for this study |

**Table A3.** Wood basic density values used to convert stem volume to biomass of taxa in the core 1 ha plot at the Warra Supersite.

| Taxon | Basic Density (kg/m$^3$) | Reference |
|---|---|---|
| *Acacia melanoxylon* | 531 | [55] |
| *Atherosperma moschatum* | 420 | [56] |
| *Dicksonia antarctica* | Biomass C = $2.70^{-3}$ $\times$ $(dbh^2 \, Ht)^{1.19}$ | [57] |
| *Eucalyptus obliqua* | 568.5 | [27] |
| Other species | 500 | Default value |

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
