# Peer review of "Measuring a Fire. The Story of the January 2019 Fire Told from Measurements at the Warra Supersite, Tasmania"

_fire, doi:10.3390/fire4020015_

Round 1
Reviewer 1 Report
Comments: One of the major events of a scientific article is missing here, which is the CONCLUSION. Therefore, it is strongly recommended to add this part with the current version in a precise manner. This part must be constructed by modifying the ‘4: discussion’ part very carefully so that ‘conclusion’ contains the brief statement of problem, analyzing method of problem statement (target or objective), major findings and future directions of research if any. However, please wait for the comments from editorial office of FIRE.
Author Response
Reviewer 1 requested that a Conclusion be inserted after the Discussion. This has been done.
Reviewer 2 Report
This is an excellent case study that will make a valuable contribution to a growing literature on fine-scale temporal trends in fuels, fire weather, fire behavior, and ecosystem responses. It serves as an imaginative example of how existing long-term ecosystem monitoring systems and protocols can be adapted to leverage their insight into active research on fire dynamics (and not just wait until a fire burns through established arrays!).
I do ask, however, that the author revisit language about carbon sequestration (thinking especially L 282-288 and any discussion based thereon).
Firstly, isn't carbon sequestered, not the CO2 molecules themselves?
And secondly, I don't recognize carbon stored in trees as sequestered. In my field, sequestration falls along a gradient of recalcitrance, and one can only really begin using the term "sequestered" for carbon in long-term pools that turn over on the scale of many centuries-millenia. Carbon in vegetation has certainly been taken out of the atmosphere and that pool is significant. But even long-lived trees are part of a relatively mobile, primarily biological pool that should not be described as sequestered carbon.
Author Response
The terminology has been amended as suggested by Reviewer 2 and now explicitly talks about "removing CO2 from the atmosphere" or "releasing CO2 into the atmosphere" (L282-286 in updated manuscript). Also made sure to change any use of the term sequestered to stored (or similar).
I also captured a suggestion by the Reviewer 2 to emphasise the opportunistic use of existing monitoring infrastructure in fire research (in the conclusion)